# Development and Implementation of a Mobile-Integrated Simulation for COVID-19 Nursing Practice: A Randomized Controlled Pretest–Posttest Experimental Design

**DOI:** 10.3390/healthcare12040419

**Published:** 2024-02-06

**Authors:** Sun-Hwa Lee, Jeong-Sil Choi

**Affiliations:** 1Shihwa Medical Center, 381-1, Siheung-si 15034, Republic of Korea; n0003@shhosp.co.kr; 2College of Nursing, Gachon University, 191 Hamhakmoero, Yeonsu-gu, Incheon 21936, Republic of Korea

**Keywords:** pandemic, nurses, high-fidelity simulation training, education, COVID-19

## Abstract

This study developed and evaluated the effects of a mobile-integrated simulation training program on infection prevention and nursing practices based on past experiences of coronavirus disease (COVID-19) care. We developed mobile videos for the experimental group and an educational booklet for the control group based on the Analysis, Design, Development, Implementation, and Evaluation (ADDIE) model. The effects of the simulation program with the use of mobile videos on knowledge of COVID-19 management, infection prevention practice confidence, and clinical decision-making anxiety and confidence were analyzed through a randomized controlled pretest–posttest experimental design. Data from 109 participants were analyzed. Five mobile videos were developed with a total duration of 43 min and 13 s. The experimental group showed significantly greater improvement in knowledge of COVID-19 management (*p* = 0.002) and infection prevention practice confidence (*p* < 0.001). Using the mobile-integrated COVID-19 nursing practice simulation program, nurses who have no experience with emerging infectious diseases can increase their infection control knowledge and infection prevention practice confidence. In conclusion, the mobile-integrated COVID-19 nursing practice simulation program was effective in increasing infection control knowledge and infection prevention practice confidence in nurses without COVID-19 care experience.

## 1. Introduction

In March 2020, the World Health Organization (WHO) declared the coronavirus disease (COVID-19) outbreak a pandemic [1]. From February 2021, the Republic of Korea began rolling out COVID-19 vaccines. However, the rate of infection continued to rise, contrary to the expectation that COVID-19 would subside with the vaccinations [2]. To accommodate the rapidly growing number of COVID-19 patients, the Korean government obligated all hospitals to designate some of their rooms for isolation in the second half of 2021. While the government provided hospitals with a support system for managing the isolation units, the responsibility for training the healthcare staff working in these units fell on the healthcare institutions [3].

Because more COVID-19 patients required treatment, the demand for nursing staff also surged. Despite having no prior experience with an infectious disease crisis, nurses assigned to COVID-19 units immediately dove into their roles to provide patient care without adequate education or training. Quality curricula to guide nurses in their practice were also unavailable [4]. Such conditions might have increased the nurses’ fear and anxiety related to the infectious disease, thus engendering psychological distress [5].

As several healthcare staff must be involved in each patient’s care in a short period of time during a pandemic, training should be short but effective. Restrictions on group training during the COVID-19 pandemic were a barrier to healthcare staff education; therefore, a new educational approach was necessary [3,4,6]. At the time, training primarily focused on donning and doffing personal protective equipment (PPE), but clinical nurses showed high demand for training in the sterilization of testing spaces, safe patient transport, disinfection of completely isolated patient routes in facilities, and actual activities performed on site [7]. However, practical education applicable to clinical settings was largely lacking owing to the absence of an established education system during a pandemic. Moreover, it is crucial to highlight that five nursing education strategies, namely critical pathways, problem-based learning, patient simulation, case-based learning, and mentorship, have consistently proven successful for graduate, undergraduate, and junior college nursing students receiving clinical nursing teaching subjects. These strategies have been identified through a systematic review and network meta-analysis [8]. In the unique context of the COVID-19 infection crisis, discovering instructional approaches that could alleviate stress and were suitable for nurses managing nurses in this scenario was paramount. Simulation education, in addition to providing the opportunity to acquire new skills, offers a secure training environment in a psychologically stable condition [9].

Considering that trained nurses’ expertise is an essential resource during a pandemic, proper training of nurses recruited for special situations, such as isolation units, minimizes risks and contributes to patient safety [6]. To ensure that healthcare professionals caring for patients in isolation are competent, a creative and specialized approach to infectious disease education is crucial to increase knowledge, improve performance, and reduce anxiety.

Based on the experiences of Korean nurses who provided patient care in restricted units while wearing PPE, we developed a mobile-integrated simulation program for COVID-19 nursing practice. We aimed to evaluate the effects of the program on knowledge on COVID-19 care, infection prevention practice confidence, and clinical decision-making anxiety and confidence.

## 2. Materials and Methods

### 2.1. Study Design

This study employed a randomized controlled pretest–posttest experimental design to investigate the effects of a mobile-integrated COVID-19 nursing practice simulation program. The program was developed based on the Analysis, Design, Development, Implementation, and Evaluation (ADDIE) model and prior COVID-19 experiences on knowledge, infection prevention practice confidence, and clinical decision-making anxiety and confidence (Figure 1).

### 2.2. Participants and Recruitment

Participants were convenience-sampled from nurses with at least 6 months of clinical experience, without experience in a COVID-19 unit, from a 500-bed secondary hospital in Gyeonggi Province, South Korea. This study was approved by the Institutional Review Board at G University (1044396-202201-HR-013-01) (6 April 2022). The sample size was determined using the G*Power 3.1.9.6 software. For an independent *t*-test (one-tailed), using a medium effect size of 0.5, significance level of 0.05, and power of 0.80, the minimum sample size was calculated as 53 for each group, for a total of 106 participants. The effect size was determined based on a similar previous study by Lee and Choi and sample design reference [10,11]. Taking potential dropouts into account, a total of 120 participants were targeted, with 60 participants in each group. Of the participants, 5 from the experimental group and 6 from the control group withdrew from the study, leaving a total of 109 participants to be included in the analysis (Figure 2).

### 2.3. Randomization and Blinding

We began developing the mobile education program and preparing the simulation environment on 1 January 2021. The simulation was created using the developed program from July to September. To minimize bias in the selection of the experimental and control groups, the groups were assigned using the RANDBETWEEN feature of Excel 2019, and the two groups were separated by shift schedule. The experimental group completed the mobile program before participating in the simulation. Nurses with prior experience in COVID-19 units and nurses who did not complete the mobile program were excluded.

### 2.4. Mobile Education Program

A mobile video was created based on COVID-19 response guidelines version 10-2 (2021), Middle East respiratory syndrome (MERS) response guidelines [12], and Ebola virus response guidelines [13]. The duration of the mobile education program was 43 min and 13 s. Five videos in the following areas were provided: basics of PPE donning and doffing, sample collection (12 min 46 s), new admission and discharge care (10 min 13 s), respiratory care (5 min 26 s), environmental management (9 min 26 s), and postmortem care (5 min 22 s). Following the completion of the simulation education, a comprehensive phase encompassing debriefing, discussion, and evaluation of educational materials and participant satisfaction was diligently conducted (Figure 3).

### 2.5. Measurements

#### 2.5.1. COVID-19 Knowledge

The COVID-19 knowledge instrument created by Taghrir et al. [14] was modified for Korean nursing students by Lee et al. [15]. With permission, a modified tool was utilized in this investigation. Besides the 15 items in the original tool, we developed and added 13 items (2 for new admission and discharge care, 3 for respiratory care, 2 for environmental management, 2 for soiled linen management, 2 for waste management, and 2 for postmortem care) based on the COVID-19 response guidelines version 10-2 [16], MERS response guidelines [12], and Ebola virus response guidelines [13], for 28 items. Subsequently, for each item, the content validity index (CVI) was computed as the number of experts giving a rating of either 3 or 4—thereby dichotomizing the ordinal scale into relevant and not relevant—divided by the total number of experts. For example, an item rated as quite or highly relevant by four of five judges would have an item-level CVI of 0.80 [17]. In the present study, content validity was evaluated by one nursing professor, one infectious disease specialist, one infection control expert with over 10 years of experience, and two infection control nurses in charge of the COVID-19 unit staff training. The CVI was 0.91. The total score ranges from 0 to 28, and a higher score indicates greater COVID-19 knowledge. Original Cronbach’s α was 0.80 in the previous study by Taghrir et al. [14], and the Kuder–Richardson (KR) 20 was 0.76 in the previous study by Lee et al. [15]. In the present study, the new validation KR was 0.60.

#### 2.5.2. Infection Prevention Practice Confidence

Infection prevention practice confidence was measured using 32 items that we developed based on the COVID-19 response guidelines (for local governments) version 10-2 developed by the Korea Disease Control and Prevention Agency [16]. The total score ranges from 32 to 160, and a higher score indicates greater confidence in infection prevention practice. The validity of the tool was evaluated by a panel of five members, including a nursing professor, an infectious disease specialist, an infection control-certified nurse, and two infection control nurses responsible for the education and management of the COVID-19 unit staff. Four of the five judges scoring an item as quite or extremely relevant would result in an item-level discrimination validity CVI of 0.80 [17]. The CVI was 0.96, and the reliability (Cronbach’s α) of the tool was 0.97 in this study.

#### 2.5.3. Clinical Decision-Making Confidence and Anxiety

The Nursing Anxiety and Self-Confidence with Clinical Decision Making (NASC-CDM^©^) tool developed by White [18] for use on United States nursing students was adapted and validated for use on nursing students in Korea by Yu et al. [19]. This 23-item tool is rated using a 6-point Likert scale, from 1 (“strongly disagree”) to 6 (“strongly agree”). The total score ranges from 23 to 138, and a higher score indicates greater clinical decision-making confidence and lower anxiety. The reliability (Cronbach’s α) of the tool was 0.94 for anxiety and 0.98 for confidence at the time of development, 0.93 for anxiety, and 0.95 for confidence in the study by Yu et al. [19], and 0.97 and 0.99 in this study, respectively.

### 2.6. High-Fidelity Simulation Training

To prepare the high-fidelity simulation room, we installed an anteroom for PPE donning and doffing, a patient room, and a debriefing room. Additionally, the equipment and supplies listed in Table 1 were prepared (Table 1).

### 2.7. Statistical Approach

The collected data were analyzed using SPSS/WIN Version 25.0 (IBM Corporation, Armonk, NY, USA). Participants’ general characteristics were analyzed with descriptive statistics. Homogeneity in general characteristics was tested using the χ²-test and Fisher’s exact test. Baseline normality of the dependent variables was analyzed using the Kolmogorov–Smirnova test. For COVID-19 knowledge that fulfilled the pre-normality assumption of dependent variables for the experimental and control groups, an independent *t*-test was used to evaluate the effectiveness of the program. Practice confidence, clinical decision-making confidence, and anxiety that failed to fulfill the normality assumption was subjected to the Mann–Whitney *U* test, a nonparametric test, to evaluate the effectiveness of the program.

## 3. Results

### 3.1. General Characteristics and Baseline Homogeneity

At baseline, the experimental and control groups did not significantly differ in sex (*p* = 0.243), marital status (*p* = 0.332), age (*p* = 0.395), clinical ladder system (CLS) (*p* = 0.460), career (*p* = 0.820), or education level (*p* = 0.996).

In terms of COVID-19-related characteristics, the groups did not significantly differ in PPE training (*p* = 0.861), COVID-19-related education (*p* = 0.106), COVID-19 care experience (*p* = 0.776), COVID-19 infection history (*p* = 0.446), cohabiting family’s history of COVID-19 infection (*p* = 0.702), or simulation training experience other than COVID-19 simulation (*p* = 0.251), confirming the homogeneity of the experimental and control groups in their general and COVID-19-related characteristics.

Both groups were also homogeneous in terms of the dependent variables at baseline, COVID-19 knowledge (*p* = 0.141), infection prevention practice confidence (*p* = 0.067), clinical decision-making anxiety (*p* = 0.696), and clinical decision-making confidence (*p* = 0.339) (Table 2).

### 3.2. Effectiveness of Mobile-Integrated COVID-19 Nursing Practice Simulation Program

The experimental group’s scores increased from the pretest to the posttest for knowledge and practice confidence. Statistically significant differences were observed between the experimental and control groups’ knowledge (t = 3.130, *p* = 0.002) and practice confidence (z = −4.175, *p* < 0.001), indicating that the mobile-integrated COVID-19 nursing practice simulation program effectively increased COVID-19 knowledge and infection prevention practice confidence.

There was no significant difference in the change of clinical decision-making anxiety score (*p* = 0.789) and clinical decision-making confidence score (*p* = 0.178) after the intervention between the two groups, suggesting that the mobile-integrated COVID-19 nursing practice simulation program was not effective in reducing clinical decision-making anxiety and improving clinical decision-making confidence (Table 3).

## 4. Discussion

This study developed and evaluated the effects of a mobile-integrated COVID-19 nursing practice simulation program based on COVID-19 unit experiences. After the onset of pandemic, each healthcare institution, recognizing the imperative need for such education in any form, developed and implemented its own educational program for COVID-19 response among healthcare professionals [20]. The early stages of COVID-19 pandemic were characterized by the uncertainty of the disease, with the future impact remaining unknown. Clinical experiences and records related to the COVID-19 pandemic during the acute phase of infectious diseases can serve as a highly valuable educational resource for future situations.

While infectious disease-prevention simulation programs have been developed for healthcare professionals since the outbreak of COVID-19 [20,21,22], they could not be implemented practically when caring for patients with infectious diseases, as the developed program was not diverse and the target audience was limited [4,20,23,24]. During the pandemic, we examined cases of applying electronic devices to infection management-related simulations, comparing them based on different situations and variables. However, comparing them across countries and devices was challenging owing to differing circumstances, and we plan to consider this aspect in future research [4,21,22,23,24].

The current program focuses on COVID-19 infection management, and the scenarios were developed on the basis of KDCA MERS, Ebola, and COVID-19 response guidelines for healthcare facilities [12,13,16]. To the best of our knowledge, this study is among the first to develop a practical program that addresses admission, discharge, environmental management, respiratory care, and end-of-life care in the isolation unit amid an infectious disease crisis. Even in the post-COVID-19 pandemic landscape, our study has enriched insights in clinical practice by addressing areas with insufficient understanding of COVID-19. While numerous studies have explored remote education methods during the pandemic, the majority have concentrated on assessing the effectiveness of existing education approaches when transitioned to remote settings. Scarcely any research has focused on providing information to nurses actively involved in clinical practice through mobile video materials, complemented by simulation to maximize competencies.

Our study was designed by consulting two previous research studies: one by Suppan et al., which used a web-based experimental group and a control group that received a PDF book; the other by Lee and Choi, which used comprehensive education in the experimental group and booklets in the control group [10,23]. After the intervention, the experimental group showed improved COVID-19 knowledge. Infection control education has been found to have positive effects on infection control knowledge, attitude, and practice, and COVID-19 infection control knowledge was a predictor of respiratory infection control practice [16,25]. During a pandemic, knowledge about the infectious disease is crucial to adhering to infection control practices [26]. In a study on the effectiveness of PPE training through an e-learning platform among pre-hospital healthcare workers during the COVID-19 pandemic, PPE selection improved in both the experimental and control groups compared with that at baseline after the intervention. However, the degree of improvement did not significantly differ between the two groups [23]. In 2014, during the occurrence of Ebola infection and death in Spain, a simulation education program was developed and implemented for the healthcare staff. Despite the provision of guidelines and protocols, it was discovered that confusion could arise during the simulation training process [27]. This suggests the need to employ various approaches in education that focuses on imparting knowledge to enhance nursing practices during an infectious outbreak. The present study integrated mobile learning into the simulation, as it minimized in-person contact during the ongoing pandemic while still being effective. Although in-person contact decreased, we strived to maintain a good quality of education by incorporating simulations, which helps improve practice. Again, this highlights the need for diverse educational approaches to enhance the effectiveness of infectious disease-related education.

The mobile-integrated COVID-19 nursing practice simulation program improved infection control practice confidence. A previous study reported that confidence in PPE donning significantly increased in the experimental group that underwent a simulation-based infection control education for isolation rooms [28]. Simulation is an effective learning modality that boosts nursing practice confidence among nurses and nursing students. Particularly, simulation for isolation rooms enables learners to acquire an array of infection control skills and gain a sense of accomplishment as they solve problems that resemble real-world problems [28]. By providing indirect experiences, establishing a simulation room and providing consistent education in the present study is believed to have contributed to boosting the nurses’ confidence in infection control practices.

Providing care for patients with infectious diseases is challenging, as evidenced by a previous report showing that even nurses are prone to making errors when required to use several types of PPE in a clinical setting [29]. Repetitive training in a high-fidelity environment is crucial to enhancing adherence to the isolation precautions during interactions with the structure of the isolation room and patient, and ensuring effective nursing activities. In addition to establishing spaces for practicums on campus, hospitals should also install simulation rooms and expand the diversity of these environments to provide consistent education and training for their nurses. In the present study, pre-learning of the isolation guidelines and infectious disease care through videos and participation in the scenario-based simulation for caring for isolated patients while wearing PPE had a positive impact on nurses’ confidence in practice. These findings highlight the need for utilizing mobile-integrated simulation programs to effectively train nurses for new infectious disease outbreaks. In particular, being able to learn about patient care without any prior experience through visual–auditory materials provided via a mobile platform would help nurses in quickly applying the learned knowledge not only during simulation but also in their practice. Therefore, simulation training using mobile learning is believed to be an effective teaching method for the care of patients with emerging infectious diseases, such as COVID-19.

Contrary to our results, a prior study reported that simulation training significantly reduced clinical decision-making anxiety [30]. Among the limited studies showing the same results on clinical decision-making anxiety, clinical experience and the complexity and uncertainty of a situation have previously been identified as predictors of clinical decision-making [31]. In the present study, complexity and uncertainty pertaining to COVID-19 are thought to have influenced the results. In the early days of the COVID-19 pandemic, lack of experience with PPE and inappropriate infection prevention education for healthcare professionals in the United Kingdom led to anxiety among medical students [32]. In this real-world setting study, one participant expressed apprehension about engaging in the nursing care of infectious disease patients during the debriefing and discussion session following the conclusion of the simulation training program. However, Aldekhyl and Arabi [22] reported no significant differences in the level of fear or anxiety after scenario-based simulation training for COVID-19. The contradictory findings between the two studies might be attributed to them being conducted at different points during the pandemic, as external factors could have influenced participants’ clinical decision-making anxiety.

Clinical decision-making confidence did not significantly differ between the experimental and control groups. In contrast, previous studies reported that the experimental and control groups significantly differed in their clinical decision-making confidence after simulation [30,33]. The present study began in July 2022, which is after the fourth Omicron wave subsided. In other words, the number of patients was declining at the time of the study, and the government reduced the number of COVID-19 isolation beds. However, an outbreak of a different variant was anticipated, and patients were allowed to be admitted to the isolation rooms in non-isolation units [3]. In terms of the time point, the fatality and infection rate from COVID-19 decreased, but many participants had already experienced COVID-19 infection personally or encountered it among their families, close acquaintances, and patients around them. Thus, clinical nurses could have been interested and motivated to engage in COVID-19 infection prevention education. That is, the control group that received an educational booklet could have studied the material on their own. These situational contexts could have influenced clinical decision-making confidence among our participants. After the conclusion of the COVID-19 nursing practice mobile-integrated education program, we conducted assessments on learning satisfaction and program evaluation. In terms of learning satisfaction, we evaluated four aspects: learner attitude, learner satisfaction, appropriateness of learning content, and learning achievement. The highest score was observed in the appropriateness of learning content, whereas the lowest evaluation was in learner attitude. Regardless of its specific form, educating the nurses remains important. Future studies should evaluate the effectiveness of education by comparing the different educational approaches with a no-education group.

A limitation of this study is that the participants were sampled from nurses at a single acute-care secondary hospital in the Republic of Korea. Thus, replication studies that consider the cultural and demographic characteristics of patients should be conducted to generalize the findings pertaining to the mobile-integrated COVID-19 nursing practice simulation program. While broad validity and reliability of the COVID-19 knowledge and practice confidence tool have been confirmed, further tool development strategies should be implemented. Additionally, comparisons with information provided through books are biased and are expected to have little impact on education. Methodological improvements have been proposed for future research, including crucial modifications to obtain more information regarding the equivalence of the two groups in terms of receiving the same education. Finally, instead of focusing on technology advancements, this study developed simulation programs based on scenarios. Future studies should practical skills be to assess the program’s efficacy.

## 5. Conclusions

The mobile-integrated COVID-19 nursing practice simulation program was effective in increasing infection control knowledge and boosting infection prevention practice confidence among nurses. Moreover, consideration of prior COVID-19 experiences allowed for the development of videos about nursing interventions and practical human resource management applicable to an infectious disease crisis. The primary goal of simulation is to prepare students or professional staff for practice in the clinical environment. The findings of this study hold significant implications for clinical practice because we explored the potential applications in real healthcare settings and considered anticipated changes and areas for improvement in nursing practice after this educational intervention. Through a practical examination of the implications, we believe that the research results will play a crucial, indirect role in providing nursing managers responsible for the education of frontline staff with valuable experiences, potentially alleviating the burden of education. The knowledge acquired through mobile videos is expected to synergize through simulations with skill enhancement for nurses caring for infectious disease patients. Additionally, we explored the potential changes in nursing practice, patient care, infection control measures, and the overall healthcare system resulting from this educational intervention. We expect that the developed videos and simulation training would help future nurses without experience with the COVID-19 pandemic gain an understanding of and be prepared to adjust to the clinical setting. Such interventions would also help them provide quality nursing care should another infectious disease crisis occur.

## Figures and Tables

**Figure 1 healthcare-12-00419-f001:**
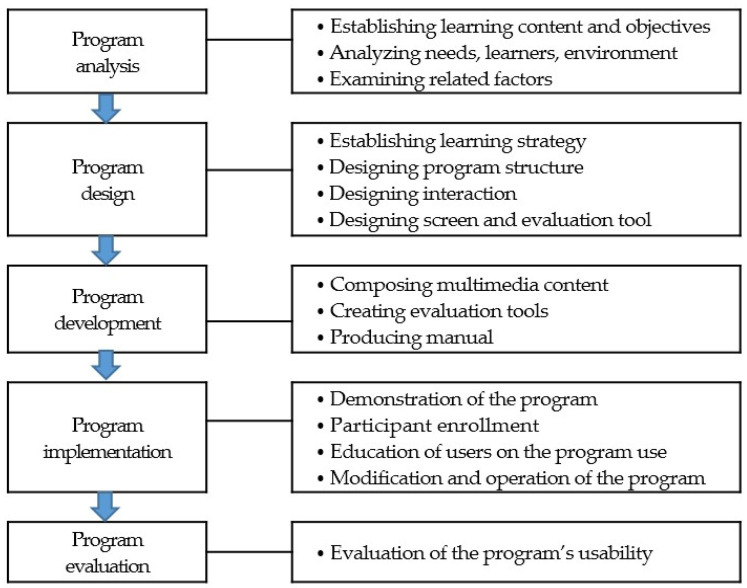
Research progress of the Analysis, Design, Development, Implementation, and Evaluation (ADDIE) model.

**Figure 2 healthcare-12-00419-f002:**
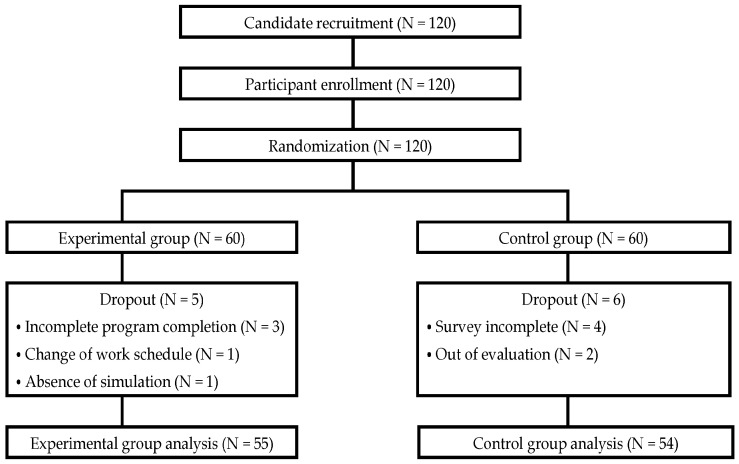
Flowchart of participant enrollment.

**Figure 3 healthcare-12-00419-f003:**
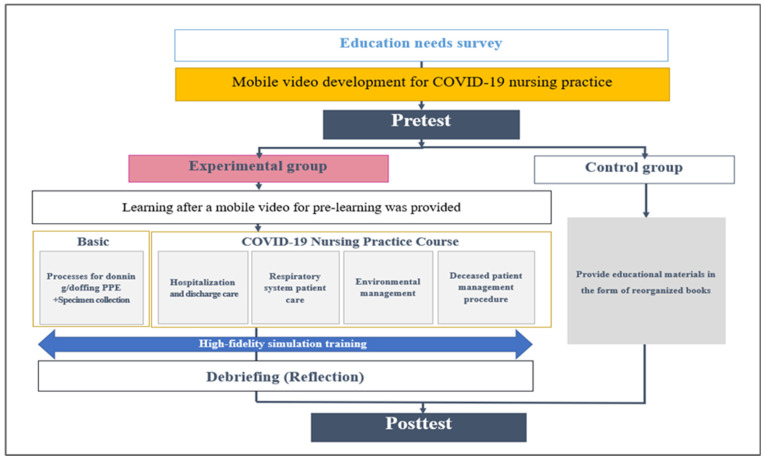
Research progress flow chart.

**Table 1 healthcare-12-00419-t001:** High-fidelity simulation training.

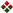 Simulation Outline
Topic	COVID-19 Care
Learning objective	Adhere to infection control guidelines when caring for COVID-19 patients admitted to the hospital.
Outline	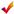 Total duration of videos (43 min 13 s) 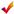 Mobile educational materials provided for pre-learning purposes 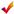 Consists of 5 sessions 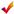 Orientation and sharing of scenarios 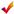 Main simulation 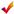 On-site debriefing and reflection 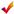 Learners: nurses with at least 6 months of clinical experience but no experience in a COVID-19 unit
Supplies	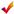 Space organization 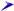 Simulation room 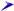 PPE donning room 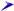 PPE doffing room 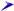 Debriefing room 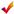 Equipment and supplies for education 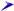 Simulator (Laerdal), body bag (corpse bag), negative pressure isolation chamber, ventilator, ventilator circuit, high-flow oxygen injector, ECG motoring machine, patient bed, disposable hospital bed cover, transparent film dressing roll, disinfecting wet wipes, gauze, medical stretcher cart, biomedical waste bin, hand sanitizer, HME ventilator filter, PPE (level D)

COVID-19: coronavirus disease, PPE: personal protective equipment, ECG: echocardiography, HME: heat and moisture exchanger.

**Table 2 healthcare-12-00419-t002:** Baseline homogeneity of participants’ general and COVID-19-related characteristics and main variables (n = 109).

Characteristic		Category	Experimental Group (*n* = 55)	Control Group (*n* = 54)	x^2^/t/Z	*p*
			*n* (%) or M ± SD	*n* (%) or M ± SD		
General characteristics						
	Sex	Male	3 (5.5)	0	3.001 *	0.243
		Female	52 (94.5)	54 (100)		
	Marital status	Single	47 (85.5)	42 (77.8)	1.072	0.332
		Married	8 (14.5)	12 (22.2)		
	CLS	None	41 (74.5)	36 (66.7)	1.546	0.460
		≥CN1	14 (25.5)	18(33.3)		
	Age	M ± SD (years)	27.74 ± 5.66	28.78 ± 5.5	−0.856	0.395
		<30 years	41 (74.5)	35 (64.8)	2.159 *	0.369
		31 to <40 years	10 (18.2)	15 (27.8)		
		≥40 years	4 (7.3)	4 (7.4)		
	Period of employment	M ± SD (months)	57.63 ± 63.39	65.41 ± 62.47	−1.044	0.301
		6 to <12 months	10 (18.2)	7 (13.0)	2.252	0.532
		1 to <2 years	15 (27.3)	10 (18.5)		
		2 to <5 years	13 (23.6)	15 (27.8)		
		≥5 years	17 (30.9)	22 (40.7)		
	Duration of current employment	M ± SD (months)	13.76 ± 5.31	13.72 ± 4.95	−0.384	0.702
		6 to <12 months	19 (34.5)	19 (35.2)	0.052	0.820
		1 to <2 years	36 (65.5)	35 (64.8)		
	Education level	Associate degree	9 (16.4)	10 (18.5))	0.002	0.996
		Bachelor’s degree or higher	46 (83.6)	44 (81.5)		
COVID-19-related characteristics						
	PPE training	Yes	42 (76.4)	42 (77.8)	0.175	0.861
		No	13 (23.6)	12 (22.2)		
	COVID-19 education	Yes	41 (74.5)	32 (59.3)	2.878	0.106
		No	14 (25.5)	22 (40.7)		
	COVID-19 care experience	Yes	44 (80)	42 (77.8)	0.081	0.776
		No	11 (11)	12 (22.2)		
	COVID-19 infection history	Yes	30 (54.5)	25 (46.3)	0.742	0.446
		No	25 (45.5)	29 (53.7)		
	Cohabiting family COVID-19 infection history	Yes	28 (50.9)	30 (55.6)	0.236	0.702
		No	27 (49.1)	24 (44.4)		
	Prior simulation other than COVID-19 simulation	Yes	22 (40)	28 (51.9)	1.541	0.251
		No	33 (60)	26 (48.1)		
Baseline homogeneity in dependent variables						
	Knowledge		19.98 ± 2.50	20.65 ± 2.59	−1.471 ^†^	0.141
	Practice confidence		110.16 ± 19.63	117.06 ± 20.01	−1.850	0.067
	Clinical decision-making anxiety		50.40 ± 20.18	49.27 ± 19.58	0.392	0.696
	Clinical decision-making confidence		74.89 ± 25.27	79.01 ± 27.20	0.957 ^†^	0.339

M: mean, SD: standard deviation, CLS: clinical ladder system, CN1: competent nursing grade 1, *: Fisher’s exact test, ^†^ Z = Mann–Whitney U test.

**Table 3 healthcare-12-00419-t003:** Effectiveness of mobile-integrated COVID-19 nursing practice simulation program.

Variable	Experimental Group (*n* = 55)			Control Group (*n* = 54)			t/Z	*p*
	Pre	Post	Difference	Pre	Post	Difference		
	M ± SD	M ± SD	M ± SD	M ± SD	M ± SD	M ± SD		
Knowledge	19.98 ± 2.50	22.88 ± 2.31	3.31 ± 3.20	20.65 ± 2.59	21.63 ± 2.39	1.33 ± 3.39	3.13	0.002
Practice	110.16 ± 19.63	130.91 ± 18.88	20.75 ± 18.36	117.05 ± 20.01	120.56 ± 21.64	3.5 ± 22.80	−4.175 *	<0.001
Clinical	50.40 ± 20.18	47.89 ± 23.56	−2.50 ± 25.45	49.27 ± 19.58	47.78 ± 22.13	−1.49 ± 25.79	−0.268 *	0.789
decision-making anxiety								
Clinical	74.89 ± 25.27	95.20 ± 24.89	20.31 ± 23.66	79.19 ± 27.43	90.46 ± 26.51	11.28 ± 34.41	−1.347 *	0.178
decision-making confidence								

M: mean, SD: standard deviation, *: Mann–Whitney U test.

## Data Availability

Data are contained within the article.

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
