# Peer review of "Development and Implementation of a Mobile-Integrated Simulation for COVID-19 Nursing Practice: A Randomized Controlled Pretest–Posttest Experimental Design"

_healthcare, 2024, doi:10.3390/healthcare12040419_

Round 1
Reviewer 1 Report
Comments and Suggestions for Authors
Dear editor and dear authors,
Thank you for the opportunity to review the manuscript “Development and implementation of a Mobile-integrated simulation for COVID-19 nursing practice: A randomized controlled pretest-posttest experimental design.”
The abstract was well-written, and the keywords were appropriate.
The introduction and discussion sections are well-framed.
The methods section needs to be clarified. I would ask to explain the constitution of the panel of experts (line 131-132).
In results, in Table 2, in the experimental group on age, we have 41 (<30 years), 9 (31 to <40 years) and 4 (≥40 years); you miss the age of one participant. On the following line, “total career,” you have on control group 7+10+16+22, equal 55, but you just have 54 participants in this group; the same happened with educational level.
In Table 3, the numbers are very close. Maybe it was an artefact of uploading onto the journal platform.
Limitations are identified. We suggested improving the conclusions. The authors should explain to readers how these results can be applied in practice and what they change after this research.
Due to the subject of study, the references are actual.
Author Response
Dear editor and dear authors,
Thank you for the opportunity to review the manuscript “Development and implementation of a Mobile-integrated simulation for COVID-19 nursing practice: A randomized controlled pretest-posttest experimental design.”
The abstract was well-written, and the keywords were appropriate.
The introduction and discussion sections are well-framed.
1)The methods section needs to be clarified. I would ask to explain the constitution of the panel of experts (line 134-138).
- Thank you for taking the time to review our manuscript. We appreciate your valuable feedback and constructive comments. We are pleased to hear that you found the abstract well-written and the keywords appropriate. Your positive feedback on the introduction and discussion sections is encouraging.
- Regarding your suggestion to clarify the methods section, specifically concerning the constitution of the panel of experts (lines 131–132), we acknowledge the need for additional information in this area. In our revision, we explained the constitution of the panel of experts. The validity of the tool was evaluated by a panel of five members, including a nursing professor, an infectious disease specialist, an infection control certified nurse, and two infection control nurses responsible for the education and management of the COVID-19unit staff.
2)In results, in Table 2, in the experimental group on age, we have 41 (<30 years), 9 (31 to <40 years) and 4 (≥40 years); you miss the age of one participant. On the following line, “total career,” you have on control group 7+10+16+22, equal 55, but you just have 54 participants in this group; the same happened with educational level.
- Thank you for your careful review and for bringing this to our attention. We have revised Table 2 as follows:
|
Age |
M±SD (years) |
7.74±5.66 |
8.78±5.5 |
-.856 |
.395 |
|
|
<30 years |
41(74.5) |
35(64.8) |
2.159* |
.369 |
||
|
31 to <40 years |
10(18.2) |
15(27.8) |
||||
|
≥40 years |
4(7.3) |
4(7.4) |
||||
|
Period of employment |
M±SD (months) |
57.63±63.39 |
65.41±62.47 |
-1.044 |
.301 |
|
|
6 to <12 months |
10(18.2) |
7(13.0) |
2.252 |
.532 |
||
|
1 to <2 years |
15(27.3) |
10(18.5) |
||||
|
2 to <5 years |
13(23.6) |
15(27.8) |
||||
|
≥5 years |
17(30.9) |
22(40.7) |
||||
|
Duration of current employment |
M±SD (months) |
13.76±5.31 |
13.72±4.95 |
-.384 |
.702 |
|
|
6 to <12 months |
19(34.5) |
19(35.2) |
.052 |
.820 |
||
|
1 to <2 years |
36(65.5) |
35(64.8) |
||||
|
Education level |
Associate degree |
9(16.4) |
10(18.5) |
.002 |
.996 |
|
|
Bachelor’s degree or higher |
46(83.6) |
44(81.5) |
3) Table 3, the numbers are very close. Maybe it was an artefact of uploading onto the journal platform.
- Thank you for pointing this out. We have accordingly changed the interval space in Table 3.
4)Limitations are identified. We suggested improving the conclusions. The authors should explain to readers how these results can be applied in practice and what they change after this research.
- Thank you for your advice. We have revised the text as follows:
Lines 364–375:
The findings of this study hold significant implications for clinical practice because we explored the potential applications in real healthcare settings and considered anticipated changes and areas for improvement in nursing practice after this educational intervention. Through a practical examination of the implications, we believe that the research results will play a crucial, indirect role in providing nursing managers responsible for the education of frontline staff with valuable experiences, potentially alleviating the burden of education. The knowledge acquired through mobile videos is expected to synergize with skills enhancement for nurses caring for infectious disease patients through simulations. Additionally, we explored the potential changes in nursing practice, patient care, infection control measures, and the overall healthcare system resulting from this educational intervention.
5) Due to the subject of study, the references are actual.
- Thank you for your positive comments on the included references.
Reviewer 2 Report
Comments and Suggestions for Authors
First of all, I would like to thank the authors for their contribution "Development and Implementation of a Mobile-Integrated Simulation for COVID-19 Nursing Practice: A Randomized Con-3 trolled Pretest-Posttest Experimental".
The study presented an important topic that would be of interest to the readership of this journal.
I encourage you to more fully illuminate your analysis process.
The explanation of the research process and the results could be explained more thoroughly to make the study more transparent and informative, as the subsequent commentary details.
I propose the following article to strengthen the methodological framework of your manuscript:
Kanaki, K., & Kalogiannakis, M. (2023). Sample design challenges: An educational research paradigm, International Journal of Technology Enhanced Learning, 15(3), 266-285.https://dx.doi.org/10.1504/IJTEL.2023.131865
On the reporting of results, the tables, and the associated data merit further attention.
The conclusions are rather short. The conclusion needs succinctly to state the intended contribution of this research, given its theoretical background, and to provide enough information to the reader about its significance.
Comments on the Quality of English LanguageIn general, the English in the present manuscript is of publication quality and requires minor improvement.
Author Response
First of all, I would like to thank the authors for their contribution "Development and Implementation of a Mobile-Integrated Simulation for COVID-19 Nursing Practice: A Randomized Con-3 trolled Pretest-Posttest Experimental".
The study presented an important topic that would be of interest to the readership of this journal.
1) I encourage you to more fully illuminate your analysis process.
- Thank you for your thoughtful review of our manuscript. We appreciate your time and valuable feedback. We are pleased to hear that you found the topic of our study to be important and of interest to the readership of the journal. Your positive comments motivate us to continue our efforts in contributing meaningful research to the field.
- We have improved our analysis process to make it more comprehensive as follows”
Lines 182–187:
The collected data were analyzed using the SPSS/WIN Version 25.0 (IBM Corporation, Armonk, NY, USA). Participants’ general characteristics were analyzed with descriptive statistics. Homogeneity in general characteristics was tested using the χ²-test and Fisher’s exact test. Baseline normality of the dependent variables was analyzed using the Kolmogorov–Smirnova test. Our hypotheses pertaining to the effectiveness of the program were tested using the independent t-test. However, non-normally distributed dependent variables were analyzed with the Mann–Whitney U test. For COVID-19 knowledge that fulfilled the pre-normality assumption of dependent variables for the experimental and control groups, an independent t-test was used to evaluate the effectiveness of the program. Practice confidence, clinical decision-making confidence, and anxiety that failed to fulfill the normality assumption was subjected to the Mann–Whitney U test, a nonparametric test, to evaluate the effectiveness of the program.
2) The explanation of the research process and the results could be explained more thoroughly to make the study more transparent and informative, as the subsequent commentary details.
- Thank you for your insightful input. Accordingly, we have added the research process and the results based on the Analysis, Design, Development, Implementation, and Evaluation (ADDIE) model.
|
Program analysis |
Establishing learning content and objectives Analyzing needs, learners, environment Examining related factors |
|
|
Program design |
Establishing learning strategy Designing program structure Designing interaction Designing screen and evaluation tool |
|
|
Program development |
Composing multimedia content Creating evaluation tools Producing manual |
|
|
Program implementation |
Demonstration of the program Participant enrollment Education of users on the program use Modification and operation of the program |
|
|
Program evaluation |
Evaluation of the program’s usability |
|
Figure 1. Research progress of the Analysis, Design, Development, Implementation, and Evaluation (ADDIE) model
3) I propose the following article to strengthen the methodological framework of your manuscript:
Kanaki, K., & Kalogiannakis, M. (2023). Sample design challenges: An educational research paradigm, International Journal of Technology Enhanced Learning, 15(3), 266-285.https://dx.doi.org/10.1504/IJTEL.2023.131865
- Thank you for the suggestion. We have revised the previous study sample size and sample design reference and added the reference you suggested. The study's merits, randomization, and blinding procedures have also been explained in detail.
- Lines 88-92:
For an independent t-test (one-tailed), using a medium effect size of 0.5, significance level of 0.05, and power of 0.80, the minimum sample size was calculated as 53 for each group, for a total of 106 participants. The effect size was determined based on a similar previous study by Lee & Choi (2022) and sample design reference [10,11]
- Lee, J.B.; Choi, J.S. Effect of an isolation-coping programme on patients isolated for colonization or infection with multi-drug-resistant organisms: a quasi-experimental study. J Hosp Infect2022,129,31-37
- Kanaki, K.; Kalogiannakis, M. Sample design challenges: an educational research paradigm. Int J Technol Enhanced Learn 2023,15,266-285. https://doi.org/10.1504/IJTEL.2023.131865
4) On the reporting of results, the tables, and the associated data merit further attention.
- We appreciate your input. We have accordingly reviewed and revised the tables and results.
5) The conclusions are rather short. The conclusion needs succinctly to state the intended contribution of this research, given its theoretical background, and to provide enough information to the reader about its significance.
- Thank you for pointing this out. We revised the conclusion section accordingly:
Lines 368–375:
Through a practical examination of the implications, we believe that the research results will play a crucial, indirect role in providing nursing managers responsible for the education of frontline staff with valuable experiences, potentially alleviating the burden of education. The knowledge acquired through mobile videos is expected to synergize with skills enhancement for nurses caring for infectious disease patients through simulations. Additionally, we explored the potential changes in nursing practice, patient care, infection control measures, and the overall healthcare system resulting from this educational intervention.
Reviewer 3 Report
Comments and Suggestions for Authors
The study aimed to evaluate the effectiveness of a mobile-integrated simulation training program in improving nurses' knowledge and confidence in managing COVID-19, particularly for those without prior experience with infectious diseases. It utilized mobile videos and an educational booklet designed using the ADDIE model. The significance of the study lies in its potential to enhance infection prevention practices and reduce clinical decision-making anxiety among healthcare professionals, which is vital during a pandemic. The results showed significant improvements, highlighting the effectiveness of digital learning tools in healthcare education.
The study concept, which involves using a mobile-integrated simulation training program to improve nursing practices in managing COVID-19, is innovative and appealing. The study's results clearly demonstrate that the application achieved its intended purpose, effectively enhancing nurses' knowledge and confidence in dealing with the pandemic. Additionally, the information presented is clear and well-structured, making it easy to understand and follow the study's findings and implications.
However, I must point out a noticeable bias in the study towards the experimental group. The design of the study seems to lack equity between the groups. The experimental group received a comprehensive learning experience that included a pre-learning session, a practice course with videos, participation in simulation training, and debriefing sessions. In contrast, the control group was only provided with materials in the form of reorganized books. This disparity in the depth and mode of learning resources provided to each group could skew the results, favoring the experimental group. Such an imbalance in the educational input challenges the fairness and validity of the comparison between the two groups, and it raises questions about the generalizability of the study's findings.
Given the disparity in resources and engagement between the experimental and control groups in the study, it raises concerns regarding the consistency in instructional design components (such as learning objectives, content, strategies, etc.) between these groups, especially when considering the ADDIE model. The distinct difference between the experimental group — which received a multifaceted approach including pre-learning sessions, practice courses with videos, simulation training, and debriefing — and the control group, which only received reorganized book materials, suggests a potential deviation from the uniform application of the ADDIE principles. This inconsistency significantly impacts the comparability of the results between the two groups. This lack of alignment in instructional strategies and materials between the groups poses a challenge in assuming equality in the educational experience and, consequently, in the validity of the study's conclusions.
Unfortunately, after carefully considering the study's methodology and the evident disparity in the treatment of the experimental and control groups, I find myself in the difficult position of recommending rejection for this study. The study held a lot of promise with its initial concept and potential impact, but it is disheartening to see it not live up to its potential. This imbalance in the learning experience compromises the integrity of the study's design and findings, making it challenging to endorse its conclusions.
Author Response
However, I must point out a noticeable bias in the study towards the experimental group. The design of the study seems to lack equity between the groups. The experimental group received a comprehensive learning experience that included a pre-learning session, a practice course with videos, participation in simulation training, and debriefing sessions. In contrast, the control group was only provided with materials in the form of reorganized books. This disparity in the depth and mode of learning resources provided to each group could skew the results, favoring the experimental group. Such an imbalance in the educational input challenges the fairness and validity of the comparison between the two groups, and it raises questions about the generalizability of the study's findings.
- Our study was designed by consulting two previous research studies: one by Suppan et al.(2020), which used a web-based experimental group and a control group that received a PDF book; the other by Lee and Choi(2022), which used comprehensive education in the experimental group and booklets in the control group. Consequently, we used these earlier studies as a guide while designing our own research and adding references.
Lines 254–257:
- Our study was designed by consulting two previous research studies: one by Suppan et al.(2020), which used a web-based experimental group and a control group that received a PDF book; the other by Lee and Choi(2022), which used comprehensive education in the experimental group and booklets in the control group
- Suppan, L.; Abbas, M.; Stuby, L.; Cottet, P.; Larribau, R.; Golay, E.; et al Effect of an e-learning module on personal protective equipment proficiency among prehospital personnel: web-based randomized controlled trial. J Med Internet Res 2020,22,e21265. https://doi.org/10.2196/21265
- Lee, J.B.; Choi, J.S. Effect of an isolation-coping programme on patients isolated for colonization or infection with multi-drug-resistant organisms: a quasi-experimental study. J Hosp Infect2022,129,31-37
Given the disparity in resources and engagement between the experimental and control groups in the study, it raises concerns regarding the consistency in instructional design components (such as learning objectives, content, strategies, etc.) between these groups, especially when considering the ADDIE model. The distinct difference between the experimental group — which received a multifaceted approach including pre-learning sessions, practice courses with videos, simulation training, and debriefing — and the control group, which only received reorganized book materials, suggests a potential deviation from the uniform application of the ADDIE principles. This inconsistency significantly impacts the comparability of the results between the two groups. This lack of alignment in instructional strategies and materials between the groups poses a challenge in assuming equality in the educational experience and, consequently, in the validity of the study's conclusions.
- Thank you. We will incorporate your suggestions as recommendations and look forward to future research in this area. Accordingly, we have added the research process and the results based on the Analysis, Design, Development, Implementation, and Evaluation (ADDIE) model.
- Lines 348–356:
While broad validity and reliability of the COVID-19 knowledge and practice confidence tool have been confirmed, further tool development strategies should be implemented. Additionally, comparisons with information provided through books are biased and are expected to have little impact on education. Methodological improvements have been proposed for future research, including crucial modifications to obtain more information regarding the equivalence of the two groups in terms of receiving the same education. Finally, instead of focusing on technology advancements, this study developed simulation programs based on scenarios. Future studies should practical skills be to assess the program's efficacy.
|
Program analysis |
Establishing learning content and objectives Analyzing needs, learners, environment Examining related factors |
|
|
Program design |
Establishing learning strategy Designing program structure Designing interaction Designing screen and evaluation tool |
|
|
Program development |
Composing multimedia content Creating evaluation tools Producing manual |
|
|
Program implementation |
Demonstration of the program Participant enrollment Education of users on the program use Modification and operation of the program |
|
|
Program evaluation |
Evaluation of the program’s usability |
|
Figure 1. Research progress of the Analysis, Design, Development, Implementation, and Evaluation (ADDIE) model
Unfortunately, after carefully considering the study's methodology and the evident disparity in the treatment of the experimental and control groups, I find myself in the difficult position of recommending rejection for this study. The study held a lot of promise with its initial concept and potential impact, but it is disheartening to see it not live up to its potential. This imbalance in the learning experience compromises the integrity of the study's design and findings, making it challenging to endorse its conclusions.
- Thank you for pointing this out. We revised the conclusion section accordingly:
- Lines 364-375:
The findings of this study hold significant implications for clinical practice because we explored the potential applications in real healthcare settings and considered anticipated changes and areas for improvement in nursing practice after this educational intervention. Through a practical examination of the implications, we believe that the research results will play a crucial, indirect role in providing nursing managers responsible for the education of frontline staff with valuable experiences, potentially alleviating the burden of education. The knowledge acquired through mobile videos is expected to synergize with skills enhancement for nurses caring for infectious disease patients through simulations. Additionally, we explored the potential changes in nursing practice, patient care, infection control measures, and the overall healthcare system resulting from this educational intervention.
- Thank you for the suggestion. We have revised the previous study sample size and sample design reference and added the reference you suggested. The study's merits, randomization, and blinding procedures have also been explained in detail.
- Lines 88-92:
For an independent t-test (one-tailed), using a medium effect size of 0.5, significance level of 0.05, and power of 0.80, the minimum sample size was calculated as 53 for each group, for a total of 106 participants. The effect size was determined based on a similar previous study by Lee & Choi (2022) and sample design reference [10,11]
- Lee, J.B.; Choi, J.S. Effect of an isolation-coping programme on patients isolated for colonization or infection with multi-drug-resistant organisms: a quasi-experimental study. J Hosp Infect2022,129,31-37
- Kanaki, K.; Kalogiannakis, M. Sample design challenges: an educational research paradigm. Int J Technol Enhanced Learn 2023,15,266-285. https://doi.org/10.1504/IJTEL.2023.131865
Reviewer 4 Report
Comments and Suggestions for Authors
Many thanks to the journal and the authors for allowing me to read their work, which is very relevant and current for science. The following are some aspects of improvement or curiosities that have come to my attention, in order to improve the work and achieve the highest quality standards.
Introduction: There is no history of the use of the research methodology (mobile training) in populations similar to the current one or the use of other training methodologies in students, which have been evaluated in depth in previous research with the use of flipped classroom, virtual reality, simulation, peer education, along with other methodologies.
What does your study contribute to the knowledge gap?
Is there a knowledge gap?
Methods: Within this section is where I find the biggest questions:
1) The sample size was calculated for a one-tailed Student's t-test, for an effect size of 0.5 points. Are the data from previous research or based on some objective aspect that ensures sufficient power? Further detail this process to ensure replicability with quality scientific evidence.
2) It is a randomized controlled trial, however, a non-probabilistic sampling procedure is used, which may affect the extrapolation capacity of the results, with a sample size of 54 and 55 participants, with respect to the minimum size of 53 participants per group, which I understand would not affect the power, however, have you considered increasing the sample size to increase its power and better evaluate the subject of the study?
3) How is it ensured that there is no contamination between the study groups? Were the videos available during the whole period, was it possible for the participants to discuss the information with each other?
4) Knowledge scale evaluated by 5 experts, why? Generally, to determine the content validity of the scales, 7-10 experts are recommended for the first evaluation. In addition, other aspects were not evaluated, such as discrimination validity, construct validity, and comprehension validity. In addition, the scale does not appear to be reliable, nor is it proven that the original validation values are not altered when new values are introduced. It is recommended to carry out a more detailed validation process of the scale, to ensure that it is appropriate for the sample.
5) The infection prevention scale is recommended as the previous one, although appropriate reliability values ​​are obtained, it cannot be guaranteed that it is valid.
Results: The results are clear and the information is presented well, complementing the tables and text.
Discussion: I also think this part can be improved.
1) It is recommended to present background information on similar research, which allows the results to be compared in a more in-depth way.
2) Compare with other countries in which similar research has been carried out, with training using electronic devices during or after the COVID-19 pandemic.
3Why has the development of skills not been evaluated with the proposed methodology?
4) I do not see a clear discussion of the results, other than a detailed analysis of the results, without going into aspects that allow improving the knowledge gap they present. It is recommended to carry out an in-depth analysis of its results in light of previous research, exposing salient aspects of both.
5) Personal reflection: Its results show an improvement in two scales that cannot be guaranteed to measure the construct they theorize and the most valid and reliable scale is the one that does not obtain a significant modification. Improving part of the methodology that allows us to know in greater depth the scope of its results, in addition, comparing an intervention with respect to giving information through books is biased and will work, because the material they give will not affect the training. Compare with some other teaching methodology (vertical, peer, etc.), the one you plan to establish as a comparator, or simply, a simulation session, with equal exposure in both groups, one with the support of mobile devices and the other only with written materials, but the two groups receive the same training, except for the study element.
Author Response
REVIWER 4COMMENT
Many thanks to the journal and the authors for allowing me to read their work, which is very relevant and current for science. The following are some aspects of improvement or curiosities that have come to my attention, in order to improve the work and achieve the highest quality standards.
Introduction: There is no history of the use of the research methodology (mobile training) in populations similar to the current one or the use of other training methodologies in students, which have been evaluated in depth in previous research with the use of flipped classroom, virtual reality, simulation, peer education, along with other methodologies.
- Thank you for your valuable comment. We have addressed this by incorporating a historical perspective on the use of the research methodology and have included relevant
- Lines 52–61:
Moreover, it is crucial to highlight that five nursing education strategies, namely critical pathways, problem-based learning, patient simulation, case-based learning, and mentorship, have consistently proven successful for graduate, undergraduate, and junior college nursing students receiving clinical nursing teaching subjects. These strategies were identified through a systematic review and network meta-analysis [8]. In the unique context of the COVID-19 infection crisis, discovering instructional approaches that can alleviate stress and are suitable for nurses managing nursing in this scenario is paramount. Simulation education, in addition to providing the opportunity to acquire new skills, offers a secure training environment in a psychologically stable condition [9].
- Ni, J.; Wu, P.; Huang, X.; Zhang, F.; You, Z.; Chang, Q.; Liao, L. Effects of five teaching methods in clinical nursing teaching: A protocol for systematic review and network meta-analysis. PLOS ONE2022,17,e0273693. https://doi.org/10.1371/journal.pone.0273693
- Hwang, W.J.; Lee, J. Effectiveness of the infectious disease (COVID-19) simulation module program on nursing students: disaster nursing scenarios. J Korean Acad Nurs2021,51,648-660. https://doi.org/10.4040/jkan.21164
What does your study contribute to the knowledge gap? Is there a knowledge gap?
- We have addressed this aspect in the discussion
Line 247-253;
- Even in the post-COVID-19 pandemic landscape, our study has enriched insights in clinical practice by addressing areas with insufficient understanding of COVID-19. While numerous studies have explored remote education methods during the pandemic, the majority have concentrated on assessing the effectiveness of existing education approaches when transitioned to remote settings. Scarcely any research has focused on providing information to nurses actively involved in clinical practice through mobile video materials, complemented by simulation to maximize competencies.
Methods: Within this section is where I find the biggest questions:
- The sample size was calculated for a one-tailed Student's t-test, for an effect size of 0.5 points. Are the data from previous research or based on some objective aspect that ensures sufficient power? Further detail this process to ensure replicability with quality scientific evidence.
- Thank you for your query. The determination of the effect size was based on a study by Lee, J. B., & Choi, J. S. [10]. Specifically, we referred to their work titled 'Effect of an isolation-coping programme on patients isolated for colonization or infection with multi-drug-resistant organisms: a quasi-experimental study,' published in the Journal of Hospital Infection, 129, 31-37. By incorporating findings from this study, we aimed to ensure methodological consistency and quality scientific evidence in our sample size calculation. The corresponding text has been provided in the manuscript as follows:
Lines 88–92:
For an independent t-test (one-tailed), using a medium effect size of 0.5, significance level of 0.05, and power of 0.80, the minimum sample size was calculated as 53 for each group, for a total of 106 participants. The effect size was determined based on a similar previous study by Lee & Choi (2022) and sample design reference [10,11].
- Lee, J.B.; Choi, J.S. Effect of an isolation-coping programme on patients isolated for colonization or infection with multi-drug-resistant organisms: a quasi-experimental study. J Hosp Infect 2022,129,31-37
- Kanaki, K.; Kalogiannakis, M. Sample design challenges: an educational research paradigm. Int J Technol Enhanced Learn 2023,15,266-285. https://doi.org/10.1504/IJTEL.2023.131865
- It is a randomized controlled trial, however, a non-probabilistic sampling procedure is used, which may affect the extrapolation capacity of the results, with a sample size of 54 and 55 participants, with respect to the minimum size of 53 participants per group, which I understand would not affect the power, however, have you considered increasing the sample size to increase its power and better evaluate the subject of the study?
- The entire sample was conveniently selected from a specific location, with subsequent recruitment conducted among nurses at the respective hospital. The allocation of participants into control and experimental groups was performed through a random process. Despite initial plans for a multi-institutional sample extraction, logistical constraints led to a more localized approach.
While contemplating an expansion of the sample size, external circumstances, particularly the ongoing challenges posed by the COVID-19 situation, prevented additional recruitment and research activities. Future endeavors will certainly consider adopting a larger sample size to enhance the statistical power of the study, allowing for a more nuanced evaluation of the research topic. We have made revisions to acknowledge the study's merits, including detailed explanations of the randomization and blinding procedures.
- Line 90-92:
- The effect size was determined based on a similar previous study by Lee & Choi (2022) and sample design reference [10,11].
- Lee, J.B.; Choi, J.S. Effect of an isolation-coping programme on patients isolated for colonization or infection with multi-drug-resistant organisms: a quasi-experimental study. J Hosp Infect2022,129,31-37
- Kanaki, K.; Kalogiannakis, M. Sample design challenges: an educational research paradigm. Int J Technol Enhanced Learn 2023,15,266-285. https://doi.org/10.1504/IJTEL.2023.131865
- How is it ensured that there is no contamination between the study groups? Were the videos available during the whole period, was it possible for the participants to discuss the information with each other?
- Ensuring the prevention of contamination between the study groups was a paramount consideration throughout the study. Random assignment led the experimental group through a sequence of survey → mobile video educational material provision → simulation → survey, whereas the control group followed a survey → distribution of pamphlets → survey → provision of mobile video educational material and simulation. This meticulous design aimed to maintain a consistent educational experience for both groups.
- Only authorized individuals have access to the mobile education system web to view the mobile educational videos.
- The mobile education videos are open for a duration of 7 days.
Line 120-122:
- Following the completion of simulation education, a comprehensive phase encompassing debriefing, discussion, and evaluation of educational materials and participant satisfaction was diligently conducted.
- Knowledge scale evaluated by 5 experts, why? Generally, to determine the content validity of the scales, 7-10 experts are recommended for the first evaluation. In addition, other aspects were not evaluated, such as discrimination validity, construct validity, and comprehension validity. In addition, the scale does not appear to be reliable, nor is it proven that the original validation values are not altered when new values are introduced. It is recommended to carry out a more detailed validation process of the scale, to ensure that it is appropriate for the sample.
- Thank you for your valuable comment. We revised CVI using a study by Polit DF, Beck CT (2006) as a basis, detailed the validation process of the scale, and added the reference.
- Lines 132–143
Subsequently, for each item, the content validity index (CVI) is computed as the number of experts giving a rating of either 3 or 4—thereby dichotomizing the ordinal scale into relevant and not relevant—divided by the total number of experts. For example, an item rated as quite or highly relevant by four of five judges would have an item-level CVI of .80 [17]. In the present study, content validity was evaluated by one nursing professor, one infectious disease specialist, one infection control expert with over 10 years of experience, and two infection control nurses in charge of the COVID-19 unit staff training. The CVI was 0.91. The total score ranges from 0 to 28, and a higher score indicates greater COVID-19 knowledge. Original Cronbach’s α was 0.80 in the previous study by Taghrir et al. [12], and the Kuder–Richardson (KR) 20 was 0.76 in the previous study by Lee et al. [13]. In the present study, the new validation KR was 0.60.
- Polit, F.; Beck, C.T. The content validity index: are you sure you know what’s being reported? Critique and recommendations. Res Nurs Health 2006,29,489-497. https://doi.org/10.1002/nur.20147
- The infection prevention scale is recommended as the previous one, although appropriate reliability values ​​are obtained, it cannot be guaranteed that it is valid.
- We have revised the CVI according to the study by Polit DF, Beck CT (2006) and added the corresponding reference.
Lines 152–156
Infection prevention practice confidence was measured using 32 items that we developed based on the COVID-19 response guidelines (for local governments) version 10-2 developed by the Korea Disease Control and Prevention Agency [14]. The total score ranges from 32 to 160, and a higher score indicates greater confidence in infection prevention practice. The validity of the tool was evaluated by a panel of five members, including a nursing professor, an infectious disease specialist, an infection control certified nurse, and two infection control nurses responsible for the education and management of the COVID-19unit staff. Four of the five judges who scored an item as quite or extremely relevant would result in an item-level discrimination validity CVI of 0.80 [17]. The CVI was 0.96, and the reliability (Cronbach’s α) of the tool was 0.97 in this study.
- Polit, F.; Beck, C.T. The content validity index: are you sure you know what’s being reported? Critique and recommendations. Res Nurs Health 2006,29,489-497. https://doi.org/10.1002/nur.20147
Results: The results are clear and the information is presented well, complementing the tables and text.
- Thank you for the positive feedback.
Discussion: I also think this part can be improved.
- It is recommended to present background information on similar research, which allows the results to be compared in a more in-depth way.
- For reference 15, a survey study on simulation centers related to COVID-19 was conducted, aligning with the context of this study. In this regard, the discussion content has been partially supplemented.
Ingrassia, P.L.; Ferrari, M.; Paganini, M.; Mormando, G. Role of health simulation centres in the COVID-19 pandemic response in Italy: a national study. BMJ Simul Technol Enhanc Learn 2021, 7, 379-384. https://doi.org/10.1136/bmjstel-2020-000813.
- Additionally, Spain, which experienced Ebola, developed simulation education related to the nursing care of Ebola patients at that time, having a similar background to this study. Therefore, the discussion content has been supplemented in this regard.
Lines 262–272;
In a study on the effectiveness of PPE training through an e-learning platform among pre-hospital healthcare workers during the COVID-19 pandemic, PPE selection improved in both the experimental and control groups compared with that at baseline after the intervention. However, the degree of improvement did not significantly differ between the two groups [23]. In 2014, during the occurrence of Ebola infection and death in Spain, a simulation education program was developed and implemented for the healthcare staff. Despite the provision of guidelines and protocols, it was discovered that confusion could arise during the simulation training process [27]. This suggests the need to employ various approaches in education that focuses on imparting knowledge to enhance nursing practices during an infectious outbreak.
- Compare with other countries in which similar research has been carried out, with training using electronic devices during or after the COVID-19 pandemic.
- Thank you for the suggestion. We revised and added references.
- line 237-241
During the pandemic, we examined cases of applying electronic devices to infection management-related simulations, comparing them based on different situations and variables. However, comparing them across countries and devices was challenging owing to the differing circumstances, and we plan to consider this aspect in future research [4,21,22,23, 24].
[4] Monesi, A.; Imbriaco, G.; Mazzoli, C.A.; Giugni, A.; Ferrari, P. In-situ simulation for intensive care nurses during the COVID-19 pandemic in Italy: advantages and challenges. Clin Simul Nurs 2022,62,52-56. https://doi.org/10.1016/j.ecns.2021.10.005
[21] Zehler, A.; Cole, B.; Arter, S. Hyflex simulation: A case study of a creative approach to unprecedented circumstances. Clin Simul Nurs 2021,60,64-68. https://doi.org/10.1016/j.ecns.2021.06.012
[22] Aldekhyl, S.S.; Arabi, Y.M. Simulation role in preparing for COVID-19. Ann Thorac Med 2020,15,134-137. https://doi.org/10.4103/atm.ATM_114_20
[23] Suppan, L.; Abbas, M.; Stuby, L.; Cottet, P.; Larribau, R.; Golay, E.; et al Effect of an e-learning module on personal protective equipment proficiency among prehospital personnel: web-based randomized controlled trial. J Med Internet Res 2020,22,e21265. https://doi.org/10.2196/21265
[24] Kasai, H.; Saito, G.; Ito, S.; Kuriyama, A.; Kawame, C.; Shikino, K.; et al COVID-19 infection control education for medical students undergoing clinical clerkship: a mixed-method approach. BMC Med Educ 2022,22,453. https://doi.org/10.1186/s12909-022-03525-1
- Why has the development of skills not been evaluated with the proposed methodology?
- This study focused on developing an educational program rather than technological advancement. We utilized easily accessible mobile devices and an education completion system using already commercialized and widely used Thus, in this study, we developed a program that incorporates simulation by planning scenarios and creating videos to nurse infectious disease patients, such as those with COVID-19. We added these aspects to the limitations of the study.
Lines 354–356;
Finally, instead of focusing on technology advancements, this study developed simulation programs based on scenarios. Future studies should practical skills be to assess the program's efficacy.
- I do not see a clear discussion of the results, other than a detailed analysis of the results, without going into aspects that allow improving the knowledge gap they present. It is recommended to carry out an in-depth analysis of its results in light of previous research, exposing salient aspects of both.
- Thank you for your insightful comments. We have revised the discussion accordingly.
Lines 262–272;
In a study on the effectiveness of PPE training through an e-learning platform among pre-hospital healthcare workers during the COVID-19 pandemic, PPE selection improved in both the experimental and control groups compared with that at baseline after the intervention. However, the degree of improvement did not significantly differ between the two groups [23]. In 2014, during the occurrence of Ebola infection and death in Spain, a simulation education program was developed and implemented for the healthcare staff. Despite the provision of guidelines and protocols, it was discovered that confusion could arise during the simulation training process [27]. This suggests the need to employ various approaches in education that focuses on imparting knowledge to enhance nursing practices during an infectious outbreak.
5) Personal reflection: Its results show an improvement in two scales that cannot be guaranteed to measure the construct they theorize and the most valid and reliable scale is the one that does not obtain a significant modification. Improving part of the methodology that allows us to know in greater depth the scope of its results, in addition, comparing an intervention with respect to giving information through books is biased and will work, because the material they give will not affect the training. Compare with some other teaching methodology (vertical, peer, etc.), the one you plan to establish as a comparator, or simply, a simulation session, with equal exposure in both groups, one with the support of mobile devices and the other only with written materials, but the two groups receive the same training, except for the study element.
- Thank you. We will incorporate your suggestions as recommendations and look forward to future research in this area.
- Line 254-257;
- Our study was designed by consulting two previous research studies: one by Suppan et al.(2020), which used a web-based experimental group and a control group that received a PDF book; the other by Lee and Choi(2022), which used comprehensive education in the experimental group and booklets in the control group. Consequently, we used these earlier studies as a guide while designing our own research and adding references.
23.Suppan, L.; Abbas, M.; Stuby, L.; Cottet, P.; Larribau, R.; Golay, E.; et al Effect of an e-learning module on personal protective equipment proficiency among prehospital personnel: web-based randomized controlled trial. J Med Internet Res 2020,22,e21265. https://doi.org/10.2196/21265
10.Lee, J.B.; Choi, J.S. Effect of an isolation-coping programme on patients isolated for colonization or infection with multi-drug-resistant organisms: a quasi-experimental study. J Hosp Infect 2022,129,31-37
- Line 348-356;
While broad validity and reliability of the COVID-19 knowledge and practice confidence tool have been confirmed, further tool development strategies should be implemented. Additionally, comparisons with information provided through books are biased and are expected to have little impact on education. Methodological improvements have been proposed for future research, including crucial modifications to obtain more information regarding the equivalence of the two groups in terms of receiving the same education. Finally, instead of focusing on technology advancements, this study developed simulation programs based on scenarios. Future studies should practical skills be to assess the program's efficacy.
Round 2
Reviewer 2 Report
Comments and Suggestions for Authors
The authors have delivered well justified revisions.
The literature review was both systematic and comprehensive.
The used methodology is adequate and appropriate.
The discussion is reasonably well laid out.
Reviewer 3 Report
Comments and Suggestions for Authors
Thank you for the explanations.
The authors considered the bias mentioned in the previous criticism as a limitation. Additionally, they presented the current situation as an alternative to situations that require urgent training, such as Covid. They supported their arguments by giving necessary references.
Reviewer 4 Report
Comments and Suggestions for Authors
The manuscript has been improved.